# Mobile phone use intention while driving among public service vehicle drivers: Magnitude and its social and cognitive determinants

Temesgen Demissie Eijigu[ID]*

Department of Psychology, Institute of Education and Behavioral Sciences, Debre Markos University, Debre Markos, Ethiopia

* Temesgen_Demissie@dmu.edu.et, temesgendem@yahoo.com

## Abstract

Despite its risks for accident and illegality, little is known about the magnitude and associated social and cognitive factors that motivate drivers to use mobile phone while driving. The present study, guided by theory of planned behavior, aimed at describing the magnitude of mobile use while driving and examining the role of attitudes, subjective norms, perceived behavioral control, and risk perceptions in predicting drivers' intentions to use mobile phone while driving. A total of 155 public service vehicle drivers, who were selected from Debre Markos Town and its vehicle terminal took part in the study. To select study participants, systematic random sampling technique was employed. The instrument used to collect data was self-report questionnaire. The results indicated that more than two-third (69%) of the participants used their mobile phone while driving over the past week. Hierarchical regression analysis displayed that perceived behavioral control, risk perception, and attitude were found to be the most significant social and cognitive predictors of public service vehicle drivers' intention to use mobile phone while driving, but not age and subjective norm variables. So as to reduce drivers' intention to use mobile phone while driving, intervention strategies should focus on enhancing their confidence to avoid this behavior; alerting drivers to the traffic control regulation and the dangers of using mobile phones while driving.

## Introduction

### Background of the study

Road traffic accidents cause about 1.24 million people death and 20 to 50 million people injury annually over the world roads [1]. Of such deaths, 92% occur in low and middle income countries that have about 53% of the world's vehicles. It has been reported that Ethiopia has one of the highest rates of road traffic fatalities per vehicle in the world. For instance, in 2007/8, 95 road traffic deaths per 10000 vehicles were registered [2]. Due to rapid population growth, increasing number of vehicles, poor road users' attitude and poor safety culture, there would

**Data Availability Statement:** All relevant data are within the paper and its Supporting Information files.

**Funding:** The author received no specific funding for this work.

**Competing interests:** The author have declared that no competing interests exist.

be a greater risk of death, injury and loss of material resources, unless measures are taken to improve road safety [2, 3].

Mobile phone use while driving is one of the factors most closely associated with traffic accidents and risky driving behaviors [4, 5]. According to the findings of scholars aforementioned, mobile phone use while driving increases a risk of a crash four times higher than those who do not use mobile phone. The risk of crash increases since it negatively affects driving performance, lane position control, and slowed reaction time [4, 6].

Despite its risks for accident, research findings from different parts of the world, for instance in Australia [7–9], New Zealand [10], Spain [11], United States [12], and Canada [13] reported a high prevalence rates of mobile phone use while driving. Similarly, a systematic review made by taking 29 articles from various countries found that there was a high frequency of both talking and texting via phone while driving [14].

In Ethiopia, local studies on magnitude and associated factors on intention to use mobile phone while driving is scant. Studies available mainly focus on the general risky driving behaviors. Whereas the frequency with which drivers had or answered calls, read or sent text messages while driving is still largely unknown. For instance, studies at Mekele city [15] and at Bahir Dar city [16] on risky driving revealed that 42.3% and 51% of drivers respectively used mobile phone while driving.

So as to discourage drivers from using mobile phone while driving, and in turn to reduce the risks of vehicle accidents and its associated fatalities, the Ethiopian government has banned mobile phone use while driving [17]. Despite legislative bans, from the researcher's everyday observation, many drivers are still using mobile phone while driving by placing themselves, passengers and pedestrians safety at risk. Likewise, review of many studies across the world [13] concluded that the use of mobile phones while driving remains quite high despite legislative efforts to limit and reduce such behavior. Understanding the magnitude of mobile phone use while driving and examining the influence of various social and cognitive factors on this behavior are indispensible in designing potential intervention strategies.

From various approaches used to understand and predict risky behaviors (e.g., mobile phone use while driving), the theory of planned behavior [TPB] has been widely employed [18, 19]. Therefore, the present study, guided by theory of planned behavior, aimed at describing the magnitude of mobile use while driving and examines the role of social and cognitive determinants. This model is a rational decision making which proposed that the immediate antecedent of behavior is intention (readiness to perform a given behavior). Intention is, in turn, the product of three social cognitive variables: attitude (i.e., the extent to which performance of the behavior is positively or negatively valued by an individual), the subjective norm (i.e., the extent to which the individual thinks that significant others would approve or disapprove of them performing the behavior), and perceived behavioral control (i.e., the individual's perception of how easy or difficult it is to perform the behavior). These determinants are themselves a function, respectively, of underlying behavioral, normative, and control beliefs [18, 19].

Theory of planned behavior has been supported by empirical evidences in successfully predicting risky driving behaviors including speeding [20], mobile phone use while driving [8, 9, 21, 22], and drink-driving [23]. For instance, studies [21, 22, 24] found that the theory of planned behavior significantly accounted for 32%, 37%, and 48% of the variance in intentions to use a mobile phone while driving respectively. Similarly, another study [8] revealed that attitude, subjective norm, perceived behavioral control and descriptive norm were accountable for 59% and 56% of variance in intention to engage in initiating and responding behaviors respectively.

However, the influence of each of the TPB variables on intention has been inconsistent. Among TPB components, perceived behavioral control remained the strongest predictor [8, 24], followed by descriptive and subjective norm, and attitude remained the weakest predictor [8]. Another study, on the other hand, indicated that attitudes consistently predicted intentions to drive while using a mobile phone followed by subjective norm, whereas perceived behavioral control did not significantly predict intention [21]. This discrepancy requires further study.

Since mobile phone while driving is risky and prevalent, assessing the role of risk perception on making decision to use mobile phone while driving seems justifiable. Accordingly, this study included risk perception as independent variable.

Some findings [13] revealed that risk perceptions were significant predictors of mobile phone use while driving. That is, the more the participants were aware of the dangers involving mobile phone use while driving, the less likely they would use mobile phone while driving. In contrast, another study [21] indicated that risk perception did not influence drivers' intentions to use their mobile phone while driving for talking. These inconsistent findings demand further research.

## Statement of the problem

Road traffic accidents in Ethiopia have been increasing and becoming the causes of significant losses of human and economic resources. The problem appears grave in Amhara regional state, where it accounted for 27.3 percent of the country's total road traffic accidents causing deaths in the year 2008/9. This fatality record is the largest among all regions compared to its low number of vehicles [25].

Consistent with Ethiopian Economic Association [EEA] report, between the years 2007/08 and 2011/12, there were 15,296 traffic accidents in Amhara region, which caused 2634 deaths, 6934 heavy injuries, and 5728 light injuries. Besides, 5994 vehicles were damaged resulting in loss of over 278 million birr [26]. Regardless of some disparities on the figure, a study in Ethiopia [27] reported that in Amhara region, between the years 2007 and 2011, a total of 10,162 road traffic accidents occurred. Both the data assured that traffic accident is a critical problem in the region.

The trend seems also similar in East Gojjam Zone [26]. For instance, in a five year period (2007/08-2011/12), there were 840 traffic accidents with an outcome of 326 deaths, 455 injuries and 330 property damages with an estimated cost of over 10.8 million birr. To see the rate of traffic accident increments, the total number of road accidents that occurred in 2007/8 was 136 while in 2012 the number was raised to 180 accidents, an increase of 75%. The actual number of traffic accidents may be greater than the official statistics because of under reporting [3].

Mobile phone use while driving is one of the major contributing factors for traffic accidents. Some empirical research reviews [1, 22, 28] reported that mobile phone use while driving causes physical and cognitive distractions. Drivers have to work on their mobile phone physically (i.e. reach, dial, hold) and operate their vehicle at the same time. Cognitively, the attention of the drivers is diverted from driving to the telephone conversations. It also impairs drivers' judgment (e.g. visual environment, lateral position) and decision making skills (e.g., slowed reaction time to traffic signals, brakes) which results in negatively impacting driving performance and could increase the risk of being involved in a road crash up to four times.

Cognizant of its consequences, the Ethiopian government has banned mobile phone use while driving. Although the country put the law into effect, the researcher's everyday observation and recent studies [16] show that mobile phone use while driving appear to be prevalent.

Despite the risks and illegality of using mobile phone while driving, little is known about the magnitude and associated social and cognitive factors that motivate drivers to use mobile phone while driving.

Therefore, this study aimed to describe the magnitude of mobile use while driving and to explore the roles of social and cognitive factors (attitude, subjective norm, perceived behavioral control and risk perception) on decision to use mobile phone while driving. This study planned to answer the following questions:

1. What is the magnitude of mobile phone use while driving among public service vehicle drivers in East Gojjam Zone?

2. Does the age of drivers predict their intention to use mobile phones while driving?

3. Do attitude, subjective norm, and perceived behavioral control predict drivers' intention to use mobile phone while driving?

4. Does risk perception predict drivers' intention to use mobile phone while driving?

## Literature review

**The magnitude of traffic accidents.**    Road traffic accidents cause about 1.24 million people death and 20 to 50 million people injury annually over the world roads with a big burden to low and middle income countries [1]. The global status report indicated that the region of Africa in general and Ethiopia, in particular has one of the highest rates of road traffic fatalities per vehicle.

Empirical studies in various countries indicated that road traffic accidents are prevalent [7, 8, 12–14, 29]. For instance, traffic accidents in Ethiopia have been increasing from time to time [30]. In 2009/10, road traffic accidents caused 2600 deaths and 7767 people injuries. After a period of three years, deaths and injuries of people increased to 3362 and 11358 respectively by 2012/13. In 2013/14, the death of people by road traffic accidents show a slightest decline (i.e., 3331) while injuries still increase (i.e., 11927). The data also depicted that the death of people per 10,000 vehicles shows a slight decline from 76.47 percent in 2009/10 to 64.18 percent in 2013/14.

Among the regions of Ethiopia, Amhara Regional State seems highly affected by traffic accidents. For instance, in 2012/13, traffic accidents caused 828 deaths and 2693 injuries which are close to one quarter of the total victims across the country [30]. Even, the third quarter report of the year 2015 indicated that traffic accidents caused 612 deaths and 2022 injuries. Similarly, the magnitude of traffic accident at East Gojjam Zone appeared to show an increasing trend [26].

Some studies [31, 32] revealed that among various types of traffic crashes, rear-end collision has become one of the leading causes of human injuries and fatalities together with massive property damage. Drivers' injury severity, one of the main concerns of the rear-end crash, is influenced by many factors related to drivers' age, gender, vehicle, crash position, airbag use, seat belt, time, two way traffic, light condition, and alcohol or drug use [32].

A similar study also investigated the differences of contributing factors to injury severity between car-strike-truck and truck-strike-car in rear-end collisions [31]. The results of their study reported that a significant difference between car-strike-truck and truck-strike-car crashes in terms of contributing factors (e.g., age group, trailing units, drinking driving, road surface and critical events) towards injury severity.

An empirical study in Mekelle Town on the prevalence and factors associated with road traffic crash depicted that 26.4 percent of taxi drivers had a road traffic crash within the past 3 years [33]. Their findings confirmed that receiving mobile phone calls while driving was reported as one of the major factors associated with road traffic crash. Since data on mobile phone use is not regularly collected when a crash occurs, the magnitude of this problem remains unknown.

**The magnitude of mobile phone use while driving.**   Although mobile phone use while driving was associated with traffic accidents, large number of drivers continue to use mobile phone while driving [7, 10, 11]. For instance, a study in Australia [7] reported that 77 percent of drivers reported using their mobile phone while driving with 40 percent daily or more and 37 percent less than once or twice a week. In 2013, a study on novice teenager drivers also concluded that 80% of participants reported making or receiving a call, and 72% reported sending or receiving a text message at least one day in a month [12].

A study in New Zealand [10] and Spain [11] reported that over 60% of participants use a mobile phone while driving. A research in Jordan also found that as mobile access to population increases, use of mobile phone while driving also increases [34]. By the same token, the number of mobile phone subscribers in Ethiopia has increased rapidly from 1.2 million in 2007 to around 23.7 million in 2013 [35] which may result an increasing number of drivers to use mobile while driving. Accordingly, few studies were conducted in Ethiopia. For instance, a study in Bahir Dar city, in north western Ethiopia [16] reported that 51 percent of drivers reported talking on mobile phone while driving and another 37 percent also used their mobile while driving but by applying a strategy of pulling over the vehicle. Similarly, a study at Mekele city reported a mobile use while driving prevalence of 42.3 percent [15]. The magnitude seems low possibly linked with the approach of data collection, i.e., interview which may result in under reporting.

The findings, altogether reported a high prevalence rates of engaging in mobile phone use while driving. The data also indicated that use of mobile phone for calling was more frequent than for text message. In support of this data, findings of previous studies [7, 11, 29] revealed that drivers are most frequently reported using their mobile phone while driving to answer calls, followed by making calls, reading text messages and sending text messages.

**The impact of mobile phone use on driving.**   Studies carried out in both natural and stimulated driving environments concluded that mobile phone use while driving negatively affect both driving performance and driving safety [10]. Use of mobile phone while driving distract drivers physically (e.g., drivers use one or both of their hands to manipulate the phone rather than the wheel), visually (e.g., move their eyes from the road to the mobile phone), auditory (e.g., shift their attention from the road environment to the sounds of the mobile phone and the conversation), and cognitively (e.g., divert their attention and thoughts from driving to the topic of the phone conversation [28]. These distractions may negatively affect drivers' performance including slower reactions and//or frequent missing to traffic signals; slower braking reactions with shorter stopping distances; reduced general awareness of other traffic; more risks in decision making like accepting shorter gaps or making fewer speed adjustments or adjustments to dangerous road conditions [28].

The overall research results suggest that the use of a mobile phone while driving negatively affects driving performance and therefore increases the crash risk. That is, drivers who use mobile phones in their vehicle have a four times higher risk of having a road crash than drivers who do not [4, 6, 11, 28].

**Determinants of mobile phone use while driving.**   To explain why drivers continue to use mobile phone while driving, underestimating the risks of mobile phone use while driving; the perceived practical, social, and psychological benefits outweigh its associated risks; social expectation to return calls immediately; social approval from significant others; personality traits and susceptibility to risk taking were cited as the possible reasons [13].

From various approaches used to understand and predict risky driving behaviors (e.g., mobile phone use while driving), the theory of planned behavior has been the most widely employed social psychological model [18, 19]. According to this model, behavior is determined by the individual's intentions to perform the behavior. The target behavior is the act that the individual is intending to perform. Intentions, in turn, are influenced by an individual's

attitude, subjective norm, and perceived behavioral control. These three TPB components are determined by an individuals' behavioral, normative, and control beliefs [19]. According to the TPB, individual will intend to use mobile phone while driving if they believe that they cannot avoid use of mobile phone; judge the specific behavior favorably; and perceive that significant others do not disapprove of mobile phone use while driving.

To begin with attitude, it reflects a person's favorable or unfavorable evaluation of performing the behavior [19]. Studies revealed that more favorable attitudes resulted in increased drivers' intention to use a mobile phone [9, 21, 22]. For instance, drivers' attitudes were significant positive predictors of drivers' willingness to use a mobile phone [22]. Likewise, empirical findings [9] found that attitude predicted intentions to both sending and reading text messages while driving. The result of another study [21] also suggest that having a more positive attitude toward mobile phone use while driving will increase the strength of intentions to use mobile phone when driving.

Next, subjective norm refers to the extent to which drivers think that significant others would approve or disapprove of them to perform or not perform the behavior [19]. A study reported that subjective norm was significant predictors of intention [21]. The result suggests that having a greater perception of normative pressure to use a mobile phone while driving will increase the strength of intentions to do so. Findings of another study [9] reported that subjective norm determined intentions to send, but not read text messages.

The third component of TPB is perceived behavioral control, which describes the driver's perceived ease or difficulty of performing or not performing the target behavior. It was found that PBC is expected to influence behavior directly or indirectly via intentions [19].

Research findings reported that drivers' perceived behavioral control significantly predict drivers' intention to use mobile phone [22]. Likewise, perceived behavioral control determined intentions to send, but not read text messages [9]. In contrast, perceived behavioral control did not influence the intentions to use a mobile phone while driving [21].

Age was the fourth variable under investigation. Studies in various countries indicated that age was significant predictor of mobile phone use while driving [11, 13, 28]. For example, a study in Spain among university workers reported that the younger drivers have used text messages more frequently than older drivers [11]. In UK, drivers under the age of 30 were almost double as likely to use a mobile phone as drivers over 30 [28]. Similarly, studies in Australia [21] found younger drivers engage in this behavior more frequently than older ones. These findings imply that younger drivers were more likely than older ones to use mobile phones while driving.

Risk perception was the final factor under investigation. Most drivers were not fully aware of the risks associated with talking on a mobile phone while driving [10]. As a result, various studies confirmed that risk perceptions were significant predictors of mobile phone use while driving [10, 13]. These findings show the negative association between risk perceptions and the use of mobile phones while driving which indicates that the more the participants were aware of the dangers involving mobile phone use while driving; the less likely they would use mobile phone while driving. In contrast, another study [21] indicated that risk perception did not influence drivers' intentions to use their mobile phone while driving for talking. These might be attributed to undermining of the risks or the benefit of using mobile phone while driving outweighs the risks [13].

## Materials and methods

### Study design

The study was cross sectional survey design, in which, it describes the existing status of mobile phone use while driving and examines the roles of social and cognitive factors on the behavior

by collecting data from large number of participants. It is assumed that social and cognitive variables (such as attitude, subjective norm, perceived behavioral control, and risk perception) and age affect the drivers' intention to use mobile phone while driving. Therefore, predictors whose effects were observed in this study were social-cognitive and demographic factors while the criterion variable would be the drivers' intention to use mobile phone while driving.

## Participants of the study

A total of 155 (152 males and 3 females) public service vehicle drivers took part in the study. Public service vehicles are motor vehicles with three or more seats (Bajaj, Minibus, Meles-tegna/44 seats/ and Bus) which are used to carry passengers. Bajaj drivers were selected from Debre Markos Town while other public service drivers were selected from Debre Markos Town vehicle terminal, which is a major route regularly used by various vehicles from all Woredas of East Gojam Zone, and towns found on the high way to Bahir Dar and Addis Ababa. All participants had ages between 20 and 55 (M = 32, SD = 6.87) with slightly more than half of them were married, had high school education and had less than 4 years driving experiences. Participants' demographic characteristics are summarized in Table 1 below.

## Population, samples and sampling technique

The populations of this study were Bajaj drivers at Debre Markos Town and public service vehicle drivers who used Debre Markos Town vehicle terminal from and to various road routes such as towns of all Woredas of East Gojjam Zone, and towns found in high way routes to Bahir Dar and Addis Ababa.

**Table 1. Participants' demographic characteristics.**

| Variables | Frequency | Percent |
|---|---|---|
| Sex | | |
| ✓ Male | 152 | 98.1 |
| ✓ Female | 3 | 1.9 |
| Age | | |
| ✓ 20–28 years | 61 | 39.4 |
| ✓ 29-34years | 38 | 24.5 |
| ✓ 35 years & above | 56 | 36.1 |
| Education level | | |
| ✓ Grade 7–8 | 23 | 14.8 |
| ✓ Grade 9–12 | 83 | 53.5 |
| ✓ Certificate | 15 | 9.7 |
| ✓ Diploma/Degree | 29 | 18.7 |
| ✓ Missing | 5 | 3.2 |
| Marital status | | |
| ✓ Never Married | 58 | 37.4 |
| ✓ Married | 83 | 53.5 |
| ✓ Separated/divorced | 14 | 9.0 |
| Driving Experiences | | |
| ✓ Less than 2.25 years | 52 | 33.5 |
| ✓ 2.30–4 years | 27 | 17.4 |
| ✓ 5–9 years | 43 | *27.7* |
| ✓ 10 years and above | 33 | 21.3 |

The total number of registered and licensed Minibuses, Melestegna and buses in East Gojjam Zone in February 2015 were 259, 170, and 7 respectively, while the total number of Bajajs at Debre Markos Town was 323. The number of buses assigned for the route of Addis Ababa and Bahir Dar to and from Debre Markos by Federal and Regional Road and Transport Bureaus varies from day to day. On average, their number was estimated to be 10–15.

Out of 774 vehicles, it was determined to select 258 sample size which is 33 percent of the population. By considering types of vehicles and their representation in the population, except buses where their number was very small, the proportion of participants was allocated to 85 Minibus, 56 Melestegna, 106 Bajaj, and 11 Bus drivers. Systematic random sampling technique was employed to select 258 participants of the study. Accordingly, a self report questionnaire was distributed to 258 participants.

From a total of 258 questionaires distributed, 156 were returned, representing a response rate of 60.47%. One participant who failed to complete the instrument appropriately was excluded from the analysis. Therefore, the analysis was made based on the data obtained from 54 Minibus drivers, 42 Melestegna, 48 Bajaj, and 11 Bus, totally 155 public service vehicle drivers.

## Procedures of data collection

Prior to data collection, the researcher requested Department of Psychology at Debre Markos University, a support letter to conduct the study. Thus, the Department granted me an official support letter to East Gojjam Road and Transport office, to Debre Markos Town Vehicle Terminal Coordinating Office, and to Debre Markos Town Bajaj Association. I made contact with those offices and explained to them the purpose of the study. After having the consent of concerned offices, I obtained the list of vehicles for sampling frame and determined the sample size.

Two data collectors were recruited, one from Debre Markos Town terminal coordinating office, who is serving as an expert, and the other from Bajaj Association who served as member of the association. Half day training was given to the data collectors on the following issues: Purpose of the study, ethical considerations, how to approach participants, and clarifications on instructions and items included in the questionnaire.

Prior to the administration of the questionnaire, the data collectors explained to participants about the purpose and its confidentiality. They also provided instructions by reading on how to fill in the questionnaire in order to avoid response biases because of poor reading and misunderstanding. Finally, the Amharic version of the questionnaire was administered to the participants. Data collections took place over a period of 8 days from February 8–16, 2015.

## Instrument

The instrument used to collect data was self-report questionnaire. The questionnaire was prepared in English and then translated into Amharic, to help the respondents clearly understand and respond questions correctly. After three experts from Debre Markos University had checked the items' clarity and relevance, minor modifications were made to all items in the way they were stated.

The questionnaire consisted of three sets of items. The first set consisted of 7 items concerning demographic characteristics (sex, age, educational level, marital status, license level, driving experience and type of vehicle).

The second part comprised of 6 items about mobile phone use while driving. It was measured by asking "Did you talk or exchange text message on a mobile phone while driving over the last week?" The response categories were "yes", "no" and "do not have mobile" scale. The

frequency of mobile phone use was measured by asking,' How often did you answer/make a call or read/ send text messages on mobile phone while driving over the last week?" with response categories ranging from 1 "never" to 4 "always." Involvement in accidents item was measured with a single item [13] on a dichotomous 'yes' and 'no' scale by asking "in the past 12 months, have you had an accident or been close to having an accident when you were using your mobile phone while driving?"

The third part included 21 items used to measure risk perception, the intention of mobile phone use while driving and social and cognitive determinants of this behavior as proposed by theory of planned behavior. The items were developed based on the guidelines for constructing questionnaire for Theory of planned behavior [36].

For each of the dimensions of this model (i.e., attitudes, subjective norms, perceived behavioral control, and intentions), and measures of risk perception, 3 to 5 items were formulated in the form of statements to be answered on a 5 point Likert scale (1 = strongly disagree, 2 = disagree, 3 = undecided, 4 = agree, 5 = strongly agree).

Out of 21 items, 4 items measured intention to use mobile phone while driving (e.g., "From now on, I plan to avoid talking on mobile phone while driving"; "I will make an effort to avoid exchanging text messages while driving."); 5 items measured perceived behavioral control (e.g., "It is hard to control myself from talking on mobile phone while driving"; I am confident that if I wanted to I could avoid talking on mobile phone while driving."); 4 items measured subjective norm (e.g., "Most of my friends and important people to me would approve of me talking on mobile phone while driving"; "Most of my friends and important people to me talk and exchange text message while driving."); 4 items measured attitude toward mobile phone use while driving (e.g., "For me, talking on mobile phone while driving is enjoyable"; For me, it is good to avoid exchanging a text message on mobile phone while driving."); and 4 items measured risk perception (e.g., "It is dangerous to exchange a text message on a mobile phone while driving"; "People who talk on a mobile phone while driving are more likely to be in a collision"; "If I talk on a mobile phone while driving, it is likely that I would be caught and fined by the police").

To test the internal reliability of each measure, Cronbach's Alpha statistics were computed. The results revealed acceptable reliability ranging from 0.65 to 0.83 (see Table 3).

## Data analysis

Data were analyzed using IBM SPSS Statistics 20. So as to describe the extent of mobile use while driving and the demographic characteristics of samples, the researcher used descriptive statistics such as percentage, mean and standard deviations. For relationship and prediction studies, various models were used in the scientific studies. For instance, in assessing geographic contributing factors to freeway local-vehicle and non-local vehicle crashes, researchers used Bayesian univariate and bivariate spatial models [37]. Besides, in prediction of freeway crash incidence across roadway characteristics and weather conditions, some other researchers used random effect model, spatial model, and spatio-temporal model [38]. However, the current study used Pearson product moment correlation to see the interrelation between the predictors and criterion variable, and multiple regressions to predict the intention of mobile phone use while driving.

Multiple regressions are a set of statistical techniques that allows one to assess the relationship between one dependent variable and several independent variables. Standard multiple regression, hierarchical regression and stepwise regression are the main analytic techniques in multiple regression [39]. For using standard multiple regression, reasons related to theory is not needed, whereas in hierarchical regression, theoretical or explicit hypothesis testing reasons are used [39].

In hierarchical regression, the researcher decides the order in which predictors enter into the equation based on logical or theoretical considerations, whereas in stepwise regression, the computer selects variables based on mathematical criterion [39, 40]. Variables selection through computer takes many key methodological decisions out of the hands of the researcher [40].

Selecting variables through computer is made based on slight differences in semi-partial correlations. Such slight statistical differences may contrast with theoretical importance. It may also leave out important predictors or may have too many predictors in the model with little contribution to predicting the dependent variable [40].

Therefore, to determine the predictors of intention to use a mobile phone while driving, the researcher used hierarchical regression analyses in three steps. This statistical technique was selected since an explicit model was tested [41]. Accordingly, predictor variables were selected in the following order: Age was entered first. Then, the theory of planned behavior variables (attitude, subjective norm, perceived behavioral control) were entered. In the third step, perception of risk variable was entered.

The current study used the following equation as a guide.

$$Y_i = (b_0 + b_1X_{i1} + b_2X_{i2} + b_3X_{i3} + b4X_{i4} + b_5X_{i5}) + \varepsilon_i \qquad [1]$$

Where Y is the outcome variable (Intention to use a mobile phone), $b_1$ is the coefficient of the first predictor (age); $b_2$ is the coefficient of the second predictor (attitude); $b_3$ is the coefficient of the third predictor (subjective norm); b4 is the coefficient of the fourth predictor (perceived behavioral control); $b_5$ is the coefficient of the fifth predictor (perceived behavioral control); and $\varepsilon$ is the difference between the predicted and the observed value of Y(the intention to use a mobile phone) for the $i$th participant.

### Ethical approval and consent to participate

Research and publication committee at Psychology Department reviewed the study proposal including the ethical procedures. The committee allowed the study to be conducted. Based on the recommendations of the committee, Department of Psychology approved the study and its procedures. Oral consent was obtained from study participants and participation was entirely based on their willingness. Participants were also informed about the purpose of the study and its procedures. Moreover, participants were informed about their right to withdraw from the study.

## Results

### The magnitude of mobile phone while driving

Table 2 presents the self-reported mobile use behavior of study participants. More than two-third (69%) of the participants used their mobile phone while driving over the past week. Of whom, 90.65% (97) answered calls, 59.81% (64) made calls, 22.43% (24) read text messages and another 5.61% (6) sent text messages.

With regard to the frequency of mobile use, 58.88%, 48.60% and 19.63% used their mobile phone rarely over the past week to answer a call, to make call or read text messages respectively. Moreover, 31.78% and 11.22% of the participants used their mobile either often or always while driving to answer or make a call over a one week period respectively.

Table 3 presents the means, standard deviations, reliabilities and correlations among each variable. Examination of the mean scores indicated that drivers held, on average, negative attitudes and intentions to use mobile phone while driving, and perceived lower social pressure

**Table 2. Percentage of participants using a mobile phone while driving.**

| Phone activities | Frequency | Percent |
|---|---|---|
| 1. Mobile phone use while driving | | |
| ✓ Yes | 107 | 69.0 |
| ✓ No | 45 | 29.0 |
| ✓ Don't have a cell phone | 3 | 1.9 |
| 1.1Answering a call | *97* | 90.65 |
| ✓ Always | 8 | 7.48 |
| ✓ Often | 26 | 24.3 |
| ✓ Rarely | 63 | 58.88 |
| ✓ Never | 10 | 9.35 |
| 1.2 Making a call | **64** | 59.81 |
| ✓ Always | 4 | 3.74 |
| ✓ Often | 8 | 7.48 |
| ✓ Rarely | 52 | 48.6 |
| ✓ Never | 43 | 40.19 |
| 1.3Reading a text message | **24** | 22.43 |
| ✓ Always | 2 | 1.87 |
| ✓ Often | 1 | 0.93 |
| ✓ Rarely | 21 | 19.63 |
| ✓ Never | 83 | 77.57 |
| 1.4 Sending a text message | 6 | 5.61 |
| ✓ Always | 2 | 1.87 |
| ✓ Often | 1 | 0.93 |
| ✓ Rarely | 3 | 1.9 |
| ✓ Never | 101 | 94.39 |

from significant others to use their mobile phone while driving with attitude, intention and subjective norm mean scores falling below the scale mid-point.

Participants also perceived that they had the ability to avoid using mobile phone while driving and that using mobile phone while driving would increase risks of accidents with both perceived behavioral control and risk perception mean scores falling above the mid-point of the scale.

Table 3 also indicates that drivers' perceived behavioral control and risk perceptions had significant negative correlations with intentions to use mobile phone while driving. Meaning, participants with high perceived ability and risk perception would have a lower intention to use mobile phone while driving. With regard to attitude, it had significant positive correlations

**Table 3. Means, standard deviations, reliabilities, and correlations among variables.**

| Variable | Mean | SD | A | 1 | 2 | 3 | 4 | 5 | 6 |
|---|---|---|---|---|---|---|---|---|---|
| Age | 32.28 | 7.02 | - | | -.03 | .002 | .05 | .03 | -.06 |
| Attitudes | 1.78 | .70 | .73 | | | .06 | -.26** | -.48** | .43** |
| Subjective norms | 2.31 | .93 | .76 | | | | -.24** | **-.15**\*\* | .09 |
| Perceived behavioral control | 3.87 | .73 | .65 | | | | | **.37**\*\* | -.61** |
| Risk perception | 4.20 | .94 | .83 | | | | | | **-.50**\*\* |
| Intention | 2.06 | .86 | .78 | | | | | | |

** Correlation is significant at the 0.01 level (2 –tailed).

with intention to use mobile phone while driving. However, age was not significantly correlated with intentions.

## Prediction of intention to use mobile phone while driving

Prior to regression analysis, some of the assumptions of regression analysis were checked. The results of statistical analysis ensured that there were no problems of collinearity and singularity. So as to determine the predictors of intention to use mobile phone while driving, three steps hierarchical multiple regression analysis was carried out. In Step 1 of the analysis, age was entered and accounted for 0.4% of the variance in drivers' intention to use a mobile phone, which was not significant, F (1, 151) = .605, p = .438.

In Step 2, the TPB variables of attitudes, subjective norms, and PBC were entered and accounted for a significant, additional 44.8% of the variance, F (4, 148) = 30.542, P = .000. Of the variables entered at step 2, perceived behavioral control (β = -.545, p = .000) and attitude (β = .289, P = .000) were significant predictors of drivers' intention to use mobile phone. This means that the drivers who held supportive attitude toward performing the behavior are more likely to hold intentions to use a mobile phone while driving. In contrast, if the drivers felt that they had the ability to avoid the behavior, they are more likely to hold weak intentions to use mobile phone.

In Step 3, risk perception was entered and accounted for a significant, additional 4% of the variance in drivers' intention to use a mobile phone, $F$ (5, 147) = 28.494, p = .000.

As Table 4 shows, the final model as a whole with all variables entered, significantly accounted for 49.2% (47.5% adjusted) of the total variance in drivers' intention to use a mobile phone, $F$ (5, 147) = 28.494, p = .000. Of which, the strongest predictors of drivers' intention to use a mobile phone was perceived behavioral control (β = -.487, p = .000), closely followed by risk perception (β = -.238, p = .000), and attitude (β = .19, p = .005). This revealed that having higher risk perception, greater perceived behavioral control, and unfavorable attitude towards

**Table 4.  Hierarchical multiple regression analysis predicting intention to use mobile phone while driving.**

| Variable | B | SEB | β | $R^2$ | Adjusted $R^2$ | $\Delta R^2$ |
|---|---|---|---|---|---|---|
| **Step 1** | | | | | | |
| Constant | 2.31 | .329 | | | | |
| Age | -.008 | .01 | -.063 | .004 | -.003 | .004 |
| **Step 2** | | | | | | |
| Constant | 4.174 | .469 | | | | |
| Age | -.003 | .007 | -.027 | | | |
| Attitude | .353 | .077 | .289** | | | |
| Subjective norm | -.055 | .058 | -.06 | | | |
| Perceived behavioral control | -.647 | .077 | -.545 ** | .452** | .437 | .448 |
| **Step 3** | | | | | | |
| Constant | 5.065 | .523 | | | | |
| Age | -.003 | .007 | .02 | | | |
| Attitude | .233 | .082 | .19 | | | |
| Subjective norm | -.069 | .056 | -.075 | | | |
| Perceived behavioral control | -.578 | .077 | -.487** | | | |
| Risk perception | -.218 | .064 | -.238** | .492** | .475 | .040 |

**p < .001
* p < .01.

mobile use while driving were associated with a weaker intention to use a mobile phone while driving.

## Discussion

The aims of the current study were to assess the magnitude of mobile phone use while driving among public service vehicle drivers and to examine the role of social and cognitive factors (attitude, subjective norm, perceived behavioral control and risk perception) on influencing the intention to use mobile phone while driving.

### The magnitude of mobile phone use while driving

Results of the current study depicted that large number of drivers (69%) reported that they did use mobile phone while driving in the previous week. Among these, over 90% used their phone while driving to answer calls, followed by making calls (59%), reading (22%), and sending (5%) text messages. These findings are consistent with other studies carried out in Ethiopia and other countries [7, 8, 12–14, 16, 29]. For instance, a study in Bahir Dar city, in north western Ethiopia [16] reported that 51 percent of drivers reported talking on mobile phone while driving and another 37 percent also used their mobile while driving but by applying a strategy of pulling over the vehicle.

A study in Australia [7] also confirmed that 77 percent of drivers reported using their mobile while driving with 40 percent daily or more and 37 percent less than once or twice a week. In contrast, a study at Mekele city, in Ethiopia [15] reported that mobile use while driving had the prevalence of 42.3%. The magnitude seems low which possibly is linked with the approach of data collection, i.e., interview which may result in under reporting.

In line with the findings of previous studies [7, 29], the current study revealed that drivers are most frequently reported using their mobile phone while driving to answer calls, followed by making calls, reading text messages and sending text messages. This data showed that people were more likely to use their mobile phone while driving for calling than text messaging. This result was also supported by a study on novice teenager drivers [12] who concluded that 80% of participants reported making or receiving a call, and 72% reported sending or receiving a text message at least one day in a month.

Moreover, in line with previous findings [8], the results of the present study showed that behaviors considered as responding behavior (answering calls & reading text messages) were performed more frequently than behaviors considered as initiating behaviors (making calls and sending text messages). The higher frequency on responding behaviors were attributed to social pressure to respond [8].

Generally, the findings of the current study revealed that the magnitude of mobile use while driving among public service vehicle drivers was found to be high. Since the behavior is illegal and risky, some drivers may not report the actual behavior. Therefore, the magnitude of mobile use while driving is expected to be greater than the self-report figure.

### Social and cognitive determinants of intention to use mobile phone while driving

Results of the current study indicated that age did not significantly predict intentions to use mobile phone while driving. This finding is inconsistent with previous studies [13, 21] of younger drivers more likely to intend to use their mobile phone while driving than their older counterparts. Since the drivers are using business vehicles, and mobile phone is currently becoming a business tool, the gap in mobile usage across age may be narrowed. In fact, such

finding is not an exception to the current study. Some researchers [22, 29] also reported that age did not significantly predict intention to use mobile phone.

The findings of the study reported that after controlling for age, TPB variables (attitudes, subjective norms, and PBC) accounted for 45.2% of the variance of intentions to use mobile phone while driving. From those variables, perceived behavioral control and attitudes were found significant predictors of drivers' intention to use mobile phone. As expected, the study also revealed that risk perception significantly accounted for drivers' intentions to use mobile phone while driving.

However, in the final model, perceived behavior control, risk perception and attitude were found to be the strongest predictors of intentions to use mobile phone.

To begin with attitude, various findings indicated that attitude predicted drivers' intention to use mobile phone [8, 15, 21, 22]. Consistent with such findings, this study demonstrated that drivers' favorable attitude toward mobile use while driving increases their intentions to do so. In relation to subjective norm, the current finding did not support the past findings which stated that drivers who perceive more approval to use their phone while driving from significant others are more likely to make decisions to do so [8, 21, 22]. Due to the risky nature of the action, significant others may not put social pressure toward drivers for using their mobile phone while driving.

Concerning perceived behavioral control, the current finding revealed that the higher perceived ability to avoid the behavior found to be significant, negative predictors of intention to use mobile phone while driving. This finding was consistent with previous findings [8, 22].

In examining the role of risk perceptions on intentions to use mobile phone, the current finding revealed that risk perception significantly contributed to the prediction of intentions to use mobile phone. This finding was consistent with previous studies [13]. This showed that the more the drivers were aware of the dangers involving mobile phone use while driving, the less likely they would use mobile phone while driving.

## Limitations and directions for future research

Although the study is significant in understanding the magnitude and the role of social and cognitive factors in influencing drivers' intention to use mobile phone while driving, the findings of the present study should also be interpreted in light of its limitations. First, the illegal nature of mobile use while driving in the country may have impacted on the accuracy of self-reporting by people who engage in this behavior. Second, the use of self-report questionnaire to assess the level of mobile use while driving may not have been an accurate measure of the actual behavior. Third, the study included only public service vehicles drivers, of which more than 98% were male drivers, may limit its generalization to female and other non-business oriented drivers.

Future research could address the limitations of the current research by, for instance, applying observation techniques and using balanced number of sexes and other non-business oriented drivers. Additionally, it may be also useful to confirm the influence of perceived behavioral control and risk perception on drivers' intention to use mobile phone.

## Conclusions

Although the Ethiopian government has banned mobile phone use while driving since 2011, the magnitude of mobile phone use while driving was quite high among public service vehicle drivers in Debre Markos town and its vicinity which may have a potential risk to themselves, passengers and pedestrians.

Partial support was found for the utility of the theory of planned behavior variable of perceived behavioral control and attitude to predict public service vehicle drivers' intentions to use mobile phone while driving, but not subjective norm significantly predicted intention.

The study results also revealed that perceived behavioral control and attitude were found to be the most significant social and cognitive predictor of intention to use mobile phone while driving, indicating that drivers with the higher perceived ability and attitude to avoid mobile use while driving were less likely to intend to use mobile phone while driving.

Additionally, the participants' perceived risk was found to be significant predictor of drivers' intentions to use mobile phone while driving, indicating that drivers with an increased awareness of the risk of mobile phone use while driving reported that they were more likely to intend to avoid using mobile phone while driving. The result of this study indicated that awareness of risks of using mobile phone while driving prevented drivers from planning to use mobile phone while driving.

The findings that public service vehicle drivers who have lower intention to use a mobile phone while driving reported greater perceived behavioral control, higher risk perception and unfavorable attitude suggests intervention with drivers should focus both on enhancing their confidence and attitude for avoiding this behavior and increasing their awareness on the dangers of using mobile phones while driving. So, those who work on the issues of drivers such as Transport Authority, Police Commission, Traffic Police Officers, and the Association of Drivers can design awareness raising programs by focusing on risk perception, attitude and perceived behavioral control. Moreover, alerting drivers about traffic control regulation and the possible consequences of using mobile phone while driving could reduce public service vehicle drivers' intention to use mobile phone while driving.

## Supporting information

**S1 Dataset.**
(SAV)

## Acknowledgments

The author would like to thank Professor Yalew Endaweke for his support in working on this topic. I would also like to acknowledge Debre Markos town terminal coordinating office and Bajaj Association for facilitating the data collection. Finally, I would like to thank all participants of this study for their time and patience in completing the questionnaire.

## Author Contributions

**Conceptualization:** Temesgen Demissie Eijigu.

**Data curation:** Temesgen Demissie Eijigu.

**Formal analysis:** Temesgen Demissie Eijigu.

**Investigation:** Temesgen Demissie Eijigu.

**Methodology:** Temesgen Demissie Eijigu.

**Project administration:** Temesgen Demissie Eijigu.

**Resources:** Temesgen Demissie Eijigu.

**Software:** Temesgen Demissie Eijigu.

**Supervision:** Temesgen Demissie Eijigu.

**Validation:** Temesgen Demissie Eijigu.

**Visualization:** Temesgen Demissie Eijigu.

**Writing – original draft:** Temesgen Demissie Eijigu.

**Writing – review & editing:** Temesgen Demissie Eijigu.

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
