## [Decision Letter · Decision Letter 0]

12 Jan 2021

PONE-D-20-37109

Mobile Phone Use Intention while Driving among Public Service Vehicle Drivers

PLOS ONE

Dear Dr. Eijigu,

Thank you for submitting your manuscript to PLOS ONE. After careful consideration, we feel that it has merit but does not fully meet PLOS ONE’s publication criteria as it currently stands. Therefore, we invite you to submit a revised version of the manuscript that addresses the points raised during the review process.

We look forward to receiving your revised manuscript.

Kind regards,

Feng Chen

Academic Editor

PLOS ONE

Journal Requirements:

Reviewers' comments:

Reviewer's Responses to Questions

**Comments to the Author**

1. Is the manuscript technically sound, and do the data support the conclusions?

Reviewer #1: Yes

Reviewer #2: Yes

2. Has the statistical analysis been performed appropriately and rigorously? 

Reviewer #1: Yes

Reviewer #2: Yes

3. Have the authors made all data underlying the findings in their manuscript fully available?

Reviewer #1: Yes

Reviewer #2: Yes

4. Is the manuscript presented in an intelligible fashion and written in standard English?

Reviewer #1: Yes

Reviewer #2: No

5. Review Comments to the Author

Reviewer #1: This paper investigates the mobile phone use while driving for drivers of public service vehicles in Ethiopia. The research topic is interesting and worth of investigation. The manuscript is well organized. Nonetheless, I would like to see the formulation of the proposed hierarchical regression model and illustrate how the hierarchical structure is considered. Moreover, a comparison between the hierarchical regression model and other alternatives is suggested to demonstrate the strength of the proposed model. Besides, it would be expected to point out the practical implications of the findings.

Reviewer #2: The current paper aims at examining the contributing factors towards mobile phone use while driving using questionnaire study. There are several issues with the paper:

1. The significance of the study should be deleted. The stated significance of the study is a bit of a stretch and does not reflect the unique contribution of the current paper.

2. The literature review is not exhaustive, the following paper is suggested to be cited and discussed in the paper:

[1] Feng Chen, Mingtao Song and Xiaoxiang Ma, Investigation on the Injury Severity of Drivers in Rear-End Collisions Between Cars Using a Random Parameters Bivariate Ordered Probit Model, International Journal of Environmental Research and Public Health, 2019, 16(14) , 2632.

[2] Shao Xiaojun, Ma Xiaoxiang, Chen Feng, Song Mingtao, Pan Xiaodong, You Kesi, 2020. A random parameters ordered probit analysis of injury severity in truck involved rear-end collisions. International journal of environmental research and public health. doi:10.3390/ijerph17020395

6. PLOS authors have the option to publish the peer review history of their article (what does this mean?). If published, this will include your full peer review and any attached files.

Reviewer #1: No

Reviewer #2: No

---

## [Author Response · Author response to Decision Letter 0]

16 Feb 2021

Responses to reviewer #1

I would like to thank the reviewer for such comments. As per the comments, I have proposed and illustrated the hierarchical regression model. Besides, standard multiple regression, hierarchical regression and stepwise regression models were compared so as to show the strength of the hierarchical regression model. Finally, in the conclusions section, I have tried to point out the practical implications of the findings. 

III. Responses to reviewer #2:

I also would like to thank the reviewer for the helpful comments. In line with the comments, I have deleted the significance of the study by pointing out its practical implications of the findings in the conclusion section. I have also discussed the empirical findings from suggested recent articles in relation to my study.

---

## [Decision Letter · Decision Letter 1]

10 Mar 2021

PONE-D-20-37109R1

Mobile phone use intention while driving among public service vehicle drivers: Magnitude, and Its social and cognitive determinants

PLOS ONE

Dear Dr. Eijigu,

Thank you for submitting your manuscript to PLOS ONE. After careful consideration, we feel that it has merit but does not fully meet PLOS ONE’s publication criteria as it currently stands. Therefore, we invite you to submit a revised version of the manuscript that addresses the points raised during the review process.

We look forward to receiving your revised manuscript.

Kind regards,

Feng Chen

Academic Editor

PLOS ONE

Journal Requirements:

Reviewers' comments:

Reviewer's Responses to Questions

**Comments to the Author**

1. If the authors have adequately addressed your comments raised in a previous round of review and you feel that this manuscript is now acceptable for publication, you may indicate that here to bypass the “Comments to the Author” section, enter your conflict of interest statement in the “Confidential to Editor” section, and submit your "Accept" recommendation.

Reviewer #1: (No Response)

Reviewer #2: All comments have been addressed

2. Is the manuscript technically sound, and do the data support the conclusions?

Reviewer #1: (No Response)

Reviewer #2: Yes

3. Has the statistical analysis been performed appropriately and rigorously? 

Reviewer #1: (No Response)

Reviewer #2: Yes

4. Have the authors made all data underlying the findings in their manuscript fully available?

Reviewer #1: (No Response)

Reviewer #2: Yes

5. Is the manuscript presented in an intelligible fashion and written in standard English?

Reviewer #1: (No Response)

Reviewer #2: Yes

6. Review Comments to the Author

Reviewer #1: Most of my comments have been addressed properly. A minor suggestion is that more references on the Pearson correlation test and hierarchical model should be cited in the Data analysis section, such as:

An empirical investigation of the factors contributing to local-vehicle and non-local-vehicle crashes on freeway. Journal of Transportation Safety and Security, 2020, DOI: 10.1080/19439962.2020.1779422.

Bayesian spatial-temporal model for the main and interaction effects of roadway and weather characteristics on freeway crash incidence. Accident Analysis and Prevention, 2019, 132, 105249.

Reviewer #2: (No Response)

7. PLOS authors have the option to publish the peer review history of their article (what does this mean?). If published, this will include your full peer review and any attached files.

Reviewer #1: No

Reviewer #2: No

---

## [Author Response · Author response to Decision Letter 1]

31 Mar 2021

I. Response to academic editor’s comment:

Thank you so much for your helpful comments. As per your comments, I have reviewed my reference list and found one omitted reference, but available in the in text citation. I have added the omitted source in the reference section, namely. East Gojjam Zone Police (2013) Traffic Accidents Statistics in East Gojjam Zone. I have also included two sources in data analysis section and in the reference list as well based upon the reviewer’s comment.

II. Responses to reviewer #1:

I would like to appreciate the reviewer for the invaluable comments. As per your comments, I have discussed the empirical findings from suggested recent articles in relation to my study.

---

## [Decision Letter · Decision Letter 2]

19 Apr 2021

Mobile phone use intention while driving among public service vehicle drivers: Magnitude, and Its social and cognitive determinants

PONE-D-20-37109R2

Dear Dr. Eijigu,

We’re pleased to inform you that your manuscript has been judged scientifically suitable for publication and will be formally accepted for publication once it meets all outstanding technical requirements.

Kind regards,

Feng Chen

Academic Editor

PLOS ONE

Additional Editor Comments (optional):

Reviewers' comments:

Reviewer's Responses to Questions

**Comments to the Author**

1. If the authors have adequately addressed your comments raised in a previous round of review and you feel that this manuscript is now acceptable for publication, you may indicate that here to bypass the “Comments to the Author” section, enter your conflict of interest statement in the “Confidential to Editor” section, and submit your "Accept" recommendation.

Reviewer #1: All comments have been addressed

Reviewer #2: All comments have been addressed

2. Is the manuscript technically sound, and do the data support the conclusions?

Reviewer #1: (No Response)

Reviewer #2: Yes

3. Has the statistical analysis been performed appropriately and rigorously? 

Reviewer #1: (No Response)

Reviewer #2: Yes

4. Have the authors made all data underlying the findings in their manuscript fully available?

Reviewer #1: (No Response)

Reviewer #2: Yes

5. Is the manuscript presented in an intelligible fashion and written in standard English?

Reviewer #1: (No Response)

Reviewer #2: Yes

6. Review Comments to the Author

Reviewer #1: (No Response)

Reviewer #2: (No Response)

7. PLOS authors have the option to publish the peer review history of their article (what does this mean?). If published, this will include your full peer review and any attached files.

Reviewer #1: No

Reviewer #2: No

---

## [Editor Report · Acceptance letter]

22 Apr 2021

PONE-D-20-37109R2 

Mobile phone use intention while driving among public service vehicle drivers:  Magnitude and its social and cognitive determinants 

Dear Dr. Eijigu:

I'm pleased to inform you that your manuscript has been deemed suitable for publication in PLOS ONE. Congratulations! Your manuscript is now with our production department. 

Kind regards, 

on behalf of

Dr. Feng Chen 

Academic Editor

PLOS ONE